# Comparison of Identification and Quantification of Polyphenolic Compounds in Skins and Seeds of Four Grape Varieties

**DOI:** 10.3390/molecules28104061

**Published:** 2023-05-12

**Authors:** Zlatina Chengolova, Yavor Ivanov, Tzonka Godjevargova

**Affiliations:** Department Biotechnology, University “Prof. Dr. A. Zlatarov”, 8010 Burgas, Bulgaria; zlatinabe4eva@abv.bg (Z.C.); qvor_burgas@abv.bg (Y.I.)

**Keywords:** grape skins, seeds, polyphenolic composition, antioxidant potential, RT-HPLC

## Abstract

The aim of this study was to identify and quantify polyphenolic compounds in skin extracts from four Bulgarian grape varieties and compare them to those of seed extracts. The values of total phenolic contents, flavonoids, anthocyanins, procyanidins and an ascorbic acid in grape skin extracts were determined. The antioxidant capacities of skin extracts were evaluated using four different methods. The total phenolic content of skin extracts was 2–3 times lower than those of seed extracts. The significant difference between total parameter values of individual grape varieties were also found. According to the total phenolic content and antioxidant capacity of skin extracts, the different grape varieties were arranged in the following order: Marselan ≥ Pinot Noir ˃ Cabernet Sauvignon ˃ Tamyanka. The individual compounds in the grape skin extracts were determined using RP-HPLC and compared with those of the seed extracts. The determined composition of skin extracts was significantly different from the seed extracts’ composition. Quantitative evaluation of the procyanidins and catechins in the skins was carried out. A correlation between phenolic contents, individual compounds and antioxidant capacity of different extracts was found. The studied grape extracts have a potential to be applied as natural antioxidants in the pharmaceutical and food industries.

## 1. Introduction

The grape is an excellent source of bioactive compounds, especially polyphenols. The content of polyphenols in the individual parts of the grape differs. The highest total phenolic content (TPC) is found in the grape seed (60–70%), followed by grape skin (28–35%) and the pulp (10%) [1,2]. The grape seeds and skins contain different polyphenols with a wide range of biological activity, which leads to neutralization of the free radicals [3]. Grape seeds and skins are waste product from the wine production, and they are a cheap source for obtaining valuable bioactive compounds.

The grape and its derivates contain flavonoids (mainly flavan-3-ols, flavonols and anthocyanins) and non-flavonoids (phenolic acids and stilbenes). Flavan-3-ols accumulate in high amounts in seeds as well as in small amounts in the skin and flesh [4]. Flavan-3-ols include monomeric compounds ((+)-catechin, (−)-epicatechin) and oligomeric compounds (mainly procyanidin B1, B2, B3, C1) synthesized through the aggregation of flavanol monomers (+)-catechin and (−)-epicatechin [5]. Flavonols (mainly quercetin, myricetin, kaempferol) accumulate considerably in grape skin and, in small quantities, can be found in the seeds of some grape varieties [6,7]. Anthocyanins are mostly concentrated in the skin of colored grape cultivars [8].

All phenolic compounds have a valuable bioactive propriety. Depending on their chemical structure, the phenolic compounds have different positive activity over human diseases [2]. There are many publications studying the individual or group act of polyphenols on different kinds of health problems [4,9,10]. The group of catechins ((+)-catechin, (−)-epicatechin and (−)-epigallocatechin-3-gallate) have antioxidative and anti-inflammatory activities [11]. There are studies about using catechins to prevent cancer and neurogenerative diseases. Alzheimer’s disease can be influenced by molecular mechanisms of those groups of compounds [12]. The catechin, epicatechin and rutin are three flavonoids that have proven antidiabetic activity. Those three antioxidants also have control over hyperglycemia [13]. Proanthocyanidins are excellent, valuable bioactive compounds that could heal diabetes, asthma and neuropathology. They could be used for the prevention of obesity, cancer and cardiovascular diseases [14]. Proanthocyanidins are used as a dietary supplement and are considered potent nutrients due to their strong antioxidant capacity [15]. The anthocyanin oenin is a pigment in the grape skins with prominent health benefits. Silva et al. [16] have described an unusual oenin antimicrobial activity. The flavonol quercetin and the flavonoid quercetin-3-glucoside have high antioxidant activity, and they are proved to have cardiovascular protective action and anticancer activity due to cytotoxic action against human cervical cancer cells [17]. The gallic acid has antibacterial properties and free radical reducing potential [18]. Also, the gallic acid could be used for protection against oxidative stress in human cells [19]. The stilbenoid polyphenol, which is resveratrol, has a high antioxidant potential and is known as a cardiovascular and neuro protector, and anticancer and antimicrobial agent [20].

The combination of monomeric, dimeric and trimeric phenolic compounds in seed and skin extracts provides a complex synergistic effect [21]. In some cases, the extracts act stronger than the pure phenolic compounds. Some authors have described that the phenolic extracts have shown higher antimicrobial potency than pure phenolic compounds [22]. In other cases, it is better to use enriched extracts with individual phenolic compounds, ensuring higher antioxidant and antimicrobial activities [23].

That is why it is important to identify and quantify the polyphenolic compounds of the grape seed and skin extracts and their correlation with antioxidant activity. This will allow us to clearly understand the extract composition and their targeted health effects. Many authors have reported on the phenolic composition and antiradical activity of wine or grape seeds, but there are few papers reporting data about grape seed and skin from the same sample [24,25,26]. There is almost no data in publications about the contents of procyanidins in skin extracts, nor discussions about the difference in the content of the procyanidins (strong antioxidants) in the skin extract and in the seed extract of the same grape variety [27,28,29,30]. The same is true regarding the content of catechin and epicatechin [31,32]. The present work addresses the flavanol composition of grape skins, comparing the results with those found from the seeds of four grape varieties; the latter information has been published in an earlier work [33].

The aim of this study was to evaluate the phenolic composition and antioxidant activity of skin extracts from four different Bulgarian grape varieties and compare them to those of seed extracts. Similarities and differences between the polyphenolic composition (especially of flavanols) of the skin and seed extracts of the different cultivars were discussed. This will allow us to evaluate their potential as a source of natural antioxidants used in the pharmaceutical and food industries.

## 2. Results and Discussion

### 2.1. Comparison of Total Phenolic Parameters of Grape Skin and Seed Extracts

The application of skin and seed extracts as sources for valuable biological active compounds is growing continuously. They possess a high antioxidant, antimicrobial and antilipid potential and can successfully be used as valuable pharmaceutical components, natural additives in foods and as dietary supplements. The identification and quantification of polyphenolic compounds in the extracts is a very important objective for the evaluation of their value. The first task in this study was to prepare the skin extracts from four Bulgarian grape varieties (Cabernet Sauvignon, Marselan, Pinot Noir and Tamyanka). The extraction was carried out using preliminary determined optimal conditions—magnetic stirring at 500 rpm, room temperature, 3 h [33]. An aqueous mixture of ethanol (70%, *v/v*) was chosen to carry out the extraction, because the ethanol is less toxic than other organic solvents. The yields of grape skin extracts were determined (Table 1). The yield was calculated as the ratio of dry extract weight to weight of dry skin weight in percent. The obtained results were compared with those of seed extracts. The yields of the skin extracts (5.5–15.42%) were found to be lower than those of the seed extracts of the four cultivars (12.03–18.34%) [33].

The values of the total phenolic contents (TPC), total flavonoids (TF), total anthocyanins (TA), procyanidins (PC) and ascorbic acid (AA) of the four skin extracts were determined and compared with analogical parameters of seed extracts (Table 2), discussed in detail in our previous paper [33]. It was found that the TPC of skin extracts are 2–2.5 times lower than those of seed extracts (Table 1). TPC values of seed extracts varied in the interval 79.06–111.22 mg GAE/g DW and, for those of skin extracts, in interval 36.28–56.17 mg GAE/g DW.

The TF values of skin extracts compared to those of seed extracts was about 10 times higher. The content of procyanidins in both extracts has the highest difference. PC content in skin extracts was lower than that of seed extracts. The skin extract of the white grape Tamyanka possess the lowest PC value. In comparison, the TA values of skin extracts were higher compared to those of seed extracts. The anthocyanin pigment content was the highest in the skin extracts of Marselan and Cabernet Sauvignon red grapes. The obtained results were logical because the TA are accumulated in red grape skins. The seeds should not contain anthocyanins, but the TA traces usually remain during the separation of seeds from skins. The AA content in the skin and seed extracts was similar. The obtained results showed that skin and seed extracts have the highest difference between TF and PC values. Many authors discussed the similar differences between the total indicator values in seed and skin extracts. Rockenbach et al. [32] indicated that the TPC of Pinot Noir seed extracts was 165.18 mg CE/g DW and the value of the skin extract was 6.60 mg CE/g DW. The TPCs in seed and skin extracts from the Cabernet Sauvignon grape were, respectively, 82.49 mg CE/g DW and 10.65 mg CE/g DW. The same authors compared the values of TF in extracts from Pinot Noir seeds and skins. They found that the TF value of the seed extract was 111.87 mg CE/g DW, but in the skin extract, the TF value was only 0.56 mg CE/g DW. In the Cabernet Sauvignon cultivar, the difference was smaller (53.12 mg CE/g DW for seed extract and 2.52 mg CE/g DW for skin extract). Guaita and Bosso [26] described that the values of TPC of seed extracts of four red grape cultivars varied from 73.7 to 107.8 mg GAE/g DW and the TPCs of skin extracts varied from 33.2 to 37.5 mg GAE/g DW. The PC values of the skin extracts were 3–5 times lower than the PC values of seed extracts. They presented the same differences between the TF values of the skin and seed extracts.

Besides that, the studied extracts from different grape cultivars presented different values of total polyphenolic indicators contained in skin and seed extracts (Table 1). It is known that the concentration of polyphenolic compounds in grapes depends on the grape cultivars and other factors, such as ripening time, climate, soil and growth location [24].

### 2.2. Validation of HPLC Method

The individual compounds in skin and seed extracts of Pinot Noir, C. Sauvignon, Marselan and Tamyanka grapes were determined using the RT-HPLC method. For this purpose, first, the calibration curves of necessary standard compounds were prepared. The absorptions of different standards were measured at 280, 320 and 360 nm (Table 2).

It is known that the differences in spectral characteristics of polyphenol compounds are linked to their chemical structures. Sovak [34] reported a review of the absorbance observations of the compound inherent for grape seed extracts. The smaller compounds with 7- and 9-C-subunits are able to absorb the light beam at wavelengths from 270 nm to 290 nm. The polyphenols with two aromatic rings and a carbon bridge in-between (like stilbenes) have maximum absorption near 310–320 nm. The next most complex compounds of this class contain three aromatic rings (flavonoids) and show a different location of the maximum 355–370 nm. Anthocyanidins and other large molecules show a peak at 500–530 nm.

The calibration graphs were constructed for every standard. The selected standards were mainly flavanols and procyanidins. The aim of this study was to prove the presence of flavanols, flavonols and procyanidins in skin extracts and to make a comparison with the individual compounds in the seed extracts. That is exactly why the absorptions of different standards were measured at 280, 320 and 360 nm. From the antocyanins was used only oenin as a standard. It is known that the skin extracts of red grapes have the antocyanins. The high TA values found in the skin extracts of red cultivars (Marselan, Cabernet Sauvignon, Pinot Noir) have proven this fact (Table 1). Gallic acid glucoside was quantified with the response factor of gallic acid given the lack of commercial standards.

The linear equations of calibration curves were calculated (Table 2). The R squared values (R^2^) for all calibration curves were very good (0.9979–0.9999). The limits of detection (LOD) and limits of quantification (LOQ) were calculated. The accuracy and repeatability of the method was determined by calculating the retention time precisions (RT). It was found that the retention time precisions were within 0.11–0.79% relative standard deviation (RSD).

### 2.3. Comparison of HPLC Identification of Phenolic Compounds in Grape Skin and Seed Extracts

The comparison between the HPLC chromatograms of skin and seed extracts from Pinot Noir and Cabernet Sauvignon, Marselan and Tamyanka grape varieties were presented on Figure 1 and Figure 2, respectively. The polarity of the polyphenolic compounds in the extracts is different. Therefore, RP-HPLC was a logical choice for extracted polyphenols identifications. The sequence of substances leaving the chromatographic column depends on the hydrophobic properties of the filling and also on the gradual increase in the concentration of acetonitrile in the mobile phase. The sequence of the eluted target compounds was presented on the chromatograms (Figure 1 and Figure 2) and in Table 3.

The composition profile of grape skin extracts was completely different from the profile of seed extracts. Analysis of soluble flavan-3-ols in grape skins is more difficult than in seeds since they are present in lower concentrations and are accompanied by other phenolic compounds such as flavonols and anthocyanins. That is why only few publications have presented the identification of flavanol composition of grape skins [27,28]. In this study, the flavonols in skins were successfully separated with the selected chromatographic conditions. Some of the phenolic compounds were found in both skin and seed extracts, like gallic acid, catechin, epicatechin, procyanidin B3 and procyanidin C1.

Comparisons made between the skin and seed chromatograms and data from Table 3 presented that the seeds are richer in monomeric flavan-3-ols: (+) catechin, (−) epicatechin, dimers: procyanidins B1, B2, B3 and trimer: procyanidin C1.

Catechin concentration in Pinot Noir, Marselan and Cabernet Sauvignon grape skins is 2, 3.5 and 6 times lower than those of grape seeds (Table 4). The concentrations of gallic acid in skin and seed extracts are similar. From procyanidins B in skin extracts was determined only procyanidin B3. Procyanidin B3 is a catechin dimer (catechin-(4α→8)-catechin). Its concentration was significant in skin extracts of Marselan and Cabernet Sauvignon red grape cultivars and equivalent to those in the seed extracts. However, the extract of Pinot Noir skins does not contain procyanidin B3. Similar results were obtained by Montealegre et al. [27]. They have found that the concentration of procyanidin B3 in skin extracts was higher with regard to concentrations of procyanidin B1 and B2. Although few in number, there are publications showing that the skin extract contains procyanidins B1 and B2 [28]. Procyanidin C1 is a procyanidin trimer consisting of three (−)-epicatechin units joined by two successive (4β→8)-links. There are no publications indicating the identification of procyanidin C1 in skin extracts. In this case, it was found that the concentration of procyanidin C1 in all skin extracts was similar to those in seed extracts. The determination of procyanidins B3 and C1 in skin extracts were very important because it is a fact that procyanidins are very strong antioxidant agents [35,36]. The (−)-epicatechin could not be located well since its peak is masked with the (−)-epigallocatechin gallate (EGCG) peak, such that their concentration was presented in the complex EC + EGCG.

The skin extracts contain compounds that were not determined in the seed extracts, like oenin, quercetin 3-gallate and quercetin. The oenin (anthocyanin) was presented in high concentrations in the skin extracts of red grapes (Marselan, Cabernet Sauvignon, Pinot Noir). The concentrations of the flavonols quercetin 3-gallate and quercetin were very low. It is known that quercetin and resveratrol are powerful antioxidants [37]. In our case, the concentration of quercetin was higher in the skin extracts from the Cabernet Sauvignon and Marselan red grapes (1.25 and 1.15 mg/g DW) compared to those of Pinot Noir and Tamyanka grapes (0.67 and 0.45 mg/g). The obtained concentrations of quercetin were higher than those obtained from other authors (0.006–1.54 mg/g DW), [28,32,38]. The concentration of resveratrol was barely detectable only in chromatogram of Marselan skin extract, but it was absent in all seed extracts. The very low resveratrol concentrations in skin extracts were found from other authors [28,32].

The tested extracts have multiple components besides that there is difference between the composition and compound concentrations in the four extracts. When determining a given component from the mixture, the additional components lead to a matrix effect. It is known that the matrix effect may cause a decrease or increase in sensitivity over time, increased baseline, imprecision of result, retention time change and chromatographic peak tailing [39,40,41]. For more precise identification of the compounds in the four extracts two approaches were used. First, the retention time precisions (RSD) were calculated for each compound from the chromatograms of each type of extract (P. Noir, Marselan, C. Sauvignon and Tamyanka) and, respectively, the combined retention time precisions from the chromatograms of the four extracts (Table 4).

It can be seen that the RSD values for each separated extract are similar to those received from chromatographic analysis of standards (Table 2). The combined retention time precisions (RSD) were higher than RSD values calculated on the base of the chromatograms of individual extracts, probably to the presence of a different matrix effect in the different four extracts. For more precise identification, a second approach was also applied through the addition of a known concentration of a standard analogous to each analyzed compound in the mixture. This standard addition leads to an increase in the absorption signal of the analyzed analyte and to prove its identity. In this way, with adding of two different concentrations from the available standards, all described compounds were separately identified in all studied skin and seed extracts. The obtained results for Marselan skin and seed extracts are presented in Table 5.

Based on these described approaches, conclusions were reached about the identification of compounds in the tested extracts and the quantity of individual components in them. Besides that, there are differences in the compound concentrations in extracts of different grape varieties. The obtained lower concentrations of individual compounds in skin extracts, compared to those of seed extracts, were correlated with the values of TPC (36.38–56.17 mg GAE/g DW), TF (2.64–6.70 mg QE/g DW) and PC (1.23–4.52 mg CE/g DW), (Table 1). According to the values of the total parameters (TPC, TF and PC) and the concentration of individual compounds, the skin extracts of different cultivars were arranged in the following order: Marselan ≥ Pinot Noir > Cabernet Sauvignon > Tamyanka.

The results we obtained for the flavanols content and quercetin of tested skin extracts are compared with the results obtained from other authors in Table 6. As we indicated above, there are few publications on the study of the flavanols content in the skin extract. It is obvious that compound concentrations vary widely in different studies. This is due to the different grape varieties, climate, growth location, extraction solvents, etc. The comparison shows that the skin extracts obtained contained a higher concentration of flavanols, catechins and quercetin compared to the analogous results obtained by other authors.

### 2.4. Comparison of Antioxidant Capacity of Skin and Seed Extracts

Antioxidant capacity is one of the most important characteristics of grape extracts, which showed the degree of scattering radicals. The obtained results for antioxidant potential of skin extracts, determined by ABTS, DPPH, FRAP and CUPRAC methods, were presented in Table 7.

All extracts possess free radical scavenging activity determined by ABTS and DPPH methods, but at different levels. High values of ABTS and DPPH antioxidant capacity compared to the results obtained with FRAP and CUPRAS methods indicated that the mechanism of the antioxidant action of extracts was carried out through hydrogen donor transfer, which could terminate the oxidation process by converting free radicals to stable forms. The obtained results were compared to the antioxidant capacity of the seed extracts. The results indicate that the skin extracts, due to the lower polyphenolic contents, had ABTS and DPPH values lower than those of the seed extracts.

FRAP and CUPRAC assays are the most widely used methods to determine the reducing power of antioxidants. These parameters indicate that the antioxidants are electron donors and can reduce the oxidized intermediates of the lipid peroxidation process. The CUPRAC assay was based on the reduction of Cu(II) to Cu(I) by antioxidants, and the FRAP assay was based on reduction of Fe(II) to Fe(I) by antioxidants.

It can be seen from Table 7 that the FRAP and CUPRAC values of skin extracts were also lower than the values of seed extracts. The similar results were presented from other authors [42,43]. Grape skins and seeds contain a complex phenolic composition. The scavenger capacity for free radicals was different among a variety of polyphenols in grape skins and seeds. Some authors presented the results for antioxidant capacity of pure phenolic compounds. Rauf et al. [14] reported that grape seed procyanidins have stronger antioxidative and anti-inflammatory effects than other polyphenols. Anastasiadi et al. [31] indicated that according to the DPPH and FRAP values, the referent phenolic compounds were arranged: quercetin > gallic acid > epicatechin > catechin > syringic acid. The identification of procyanidins, catechin, quercetin and quercetin 3-gallate in studied grape skin extracts showed that they possess compounds with high antioxidant activity. All obtained results indicated that the skin and seed extracts from red cultivars Marselan, Cabernet Sauvignon and Pinot Noir had significantly higher antioxidant capacity than those of white cultivar Tamyanka. The obtained antioxidant capacities correlated very well with the polyphenolic content in grape skin and seed extracts.

The obtained grape extracts can be successfully used as valuable sources of biologically active substances with high antioxidant potential. The determined bioactive com-pounds are rather important due to their positive effect on civilizational diseases. Catechins have good anti-diabetic, anti-obesity, anti-infectious and hepatoprotective properties [12,13]. Gallic acid is well known for its antioxidant, antiapoptotic, cardioprotective, neuroprotective, antiproliferative and anti-cancer proprieties [18,19]. Quercetin and quercetin 3-gallate are strong antioxidants, inhibit human platelet aggregation in vitro and exhibit potential anticancer properties [37,38]. Procyanidins are demonstrated antioxidant, antibacterial, anti-viral, anticarcinogenic, anti-inflammatory, anti-allergic and vasodilatory actions [14,36].

## 3. Materials and Methods

### 3.1. Materials

The materials used in this study were the wastes (pomaces) from the vinification of three red wines *Vitis vinifera* L. cv. Pinot Noir, C. Sauvignon, Marselan grapes and one white wine *Viis vinifera* L. cv. Tamyanka grape (Pink Pelikan Winery Ltd., Silistra, Bulgaria). These grape sorts are grown in the Danube region, near the city of Ruse, Bulgaria. Marselan is a recent crossing between two famous red grape varieties, *Vitis vinifera* L. cv. Grenache and Cabernet Sauvignon. Tamyanka is a French Muscat.

The chemicals used for the experiment were ethanol 99.9% *v/v* (Valerus, Sofia, Bulgaria), sodium carbonate (>99%), methanol, vanillin, ascorbic acid, 2,2-diphenyl-1-picrylhydrazyl (DPPH), 6-hydroxy-2,5,7,8-tetramethylchroman-2-carbohyllic acid (Trolox 97%), 2,2-azino-bis (3-ethylbenzotiazoline-6-sulphonic acid) (ABTS), potassium persulfate, 2N solution Folin-Ciocalteu reagent, gallic acid, quercetin, (+) catechin, sodium hydroxide, copper chloride, neocuproine, sodium acetate, ammonium acetate, tripyridyltriazine (TPTZ), ferric chloride, cupric chloride, trichloroacetic acid, butylated hydroxytoluene (BHT) and thiobarbituric acid purchased from Sigma-Aldrich Co., Steinheim am Albuch, Germany. The reagents sodium nitrite and aluminum chloride hexahydrate were purchased from Merck Co., Darmstadt, Germany. Deionized water purified by ELGA’s water purification systems (UK) was used throughout the experiments. The reagents for HPLC analysis, as phosphoric acid, acetonitrile, (+)-catechin, (−)-epicatechin, (−)-epigallocatechin gallate, procyanidin B1, procyanidin B2, procyanidin B3, procyanidin C1, oenin chloride, quercetin 3-glucoside, resveratrol, quercetin and rutin were purchased by Sigma-Aldrich (Burghausen, Germany), and they were HPLC-grade.

### 3.2. Preparation of Grape Skin and Seed Extracts

Skins were separated from the seeds by rubbing the mixture over a plastic sieve. Then the treatment of grape seeds and skins was carried out separately. Processing included separately washing, drying at 40 °C for 14 h and storing at 4 °C. For each experiment, a certain amount of dry grape seeds or skins was grinded to powder with a diameter of 2.5–22.5 μm. The mixture of grape skin powder (5 g) with 25 mL 70% aqueous ethanol was stirred on a magnetic stirrer MMS-3000 (Boeco, Hamburg, Germany) at a constant stirring rate of 500 rpm, at ambient temperature and pressure, for 3 h. The supernatant was separated, and the process was repeated with another 25 mL 70% aqueous ethanol. The resulting mixture was centrifuged at 4830× *g* for 10 min. The supernatant was separated and concentrated to 1 mL in a vacuum evaporator (Rotavapor R-215, Buchi, Flawil, Switzerland) in a water bath at 50–60 °C and 100–175 hPa vacuum pressure. The extract of grape seeds was prepared at the same conditions [33].

### 3.3. Spectrophotometric Methods for Determination of Skin and Seed Polyphenols

The methods for determining the total phenolic content (TPC), total flavonoids (TF), procyanidins (PC), total anthocyanin pigments (TA) and ascorbic acid content (AA) were described in detail in our previous manuscript [33]. Absorbance was measured at spectrophotometer 6900 UV-Vis JENWAY, Colmworth, UK.

Total phenolic content was determined by the Folin–Ciocalteu assay [33]. The absorbances were measured at 725 nm. The standard curve was obtained on the basis of gallic acid with a concentration of 50–450 µg/mL in ethanol. Total polyphenolic content was expressed as milligram-equivalent gallic acid per gram of dry weight matter, mg GAE/g DW.

Total flavonoids were determined using the aluminum complexation assay [33]. The absorbances were measured spectrophotometrically at 510 nm. The standard curve was obtained on the basis of a methanol solution of quercetin with a concentration of 0–1 mg/mL. Total flavonoids were expressed as milligram-equivalent quercetin per gram of dry weight matter, mg QE/g DW.

Procyanidins were estimated according to the procedure described by Brezoiu et al. [44]. An extract solution with a concentration of 300 µg/mL in methanol was prepared. A calibration curve was prepared using absorption for catechin solutions with concentrations ranging from 20 to 500 µg/mL. Different mixtures were prepared and incubated for 20 min at 30 °C and then their absorption was measured at 500 nm. Procyanidin content was expressed as mg (+)-catechin per gram of dry weight matter, mg CE/g DW.

The content of total anthocyanin pigments was determined according to the method described by Brezoiu et al. [44]. The extract (250 µL) was diluted with buffer solution at pH 1 to obtain an absorbance of 1.1. The dilution factor was determined. Then, the extract was diluted with buffer solution at pH 4.5 using the same dilution factor. The absorbance was measured at 520 nm. The concentration of anthocyanin pigments was expressed as cyanidin-3-glucoside equivalents (mg/L).

The ascorbic acid content was determined according to the methodology described by Brezoui et al. [44]. Extract (500 µL) was mixed with 250 µL of a Folin–Ciocalteu reagent that was previously diluted with 4.5 mL ultrapure water. The absorbance at 765 nm was measured. The standard curve was obtained on the basis of ascorbic acid in concentrations of 10–500 μg/mL in ethanol.

### 3.4. Spectrophotometric Methods for Determination of Antioxidant Capacity of Skin and Seed Extracts

The ABTS assay was carried out as described by Sofi et al. [45]. The ABTS radical was prepared by mixing equimolar amounts of 2.6 mM of potassium persulfate and 7.4 mM ABTS, followed by 16 h incubation in a dark place, at room temperature. Before analysis, the 1 mL ABTS+• solution was diluted with 60 mL methanol to an absorbance of 1.1 ± 0.02 at 734 nm. Samples (100 µL) were allowed to react with 2 mL of ABTS+• solution for 10 min, in a dark place. Afterwards, the solution absorbance at 734 nm was measured. A standard line is drawn in the linear interval 25–600 µM Trolox (TE). Results were expressed as µM TE equivalents/g dry weight matter.

Determination of the DPPH radical scavenging activity was performed as described by Sofi et al. [45]. The sample (100 µL) was mixed with 2 mL of daily prepared DPPH• solution (24 mg in 100 mL methanol) for 20 min at room temperature. The decrease in absorbance was red spectrophotometrically at 515 nm. A linear regression for the Trolox standards (25–800 µM) was constructed. Results were expressed as µM TE equivalents/g dry weight matter.

Ferric reducing antioxidant potential (FRAP) assay was performed based on the method of Benzie and Strain [46]. The FRAP reagents were prepared daily by mixing 10 volumes of 300 mM sodium acetate buffer (pH 3.6) with 1 volume of 10 mM TPTZ solution and 1 volume 20 mM ferric chloride. Samples (150 μL) were allowed to react with FRAP reagent (2.85 mL) for 30 min in dark condition. The absorption was measured at 539 nm. A standard line is drawn in the linear interval from 25 to 800 µM Trolox. FRAP values were expressed as μmol TE per g dry matter.

The cupric reducing antioxidant capacity (CUPRAC) of the grape seed and skin extracts was determined according to the method of Apak et al. [47]. An amount of 1 mL 7.5 mM neocuprine and 1 mL 1M ammonium acetate buffer (pH 7.0) solutions were added to a test tube with 1mL 10 mM Cu(II). Extracts were added to the initial mixture so as to make the final volume of 4.1 mL. The tubes were stoppered and the absorbance at 450 nm was recorded against a reagent blank after 30 min. Results were expressed as μmol TE per g dry matter.

### 3.5. HPLC Assays for Determination of Phenolic Composition of Skin and Seed Extracts

The antioxidant content of grape extracts was determined using HPLC (Varian 920-LC Liquid Chromatograph, Sudbury, UK) and reverse phase column (Stable Bond analytical column Zorbax SB-C18: 5 μm, 4.6 × 250 mm) with guard column (Zorbax SB-C18: 5 μm, 4.6 × 12.5 mm). The instrument was equipped with a photo diode array (PDA) detector and set range 200–600 nm, single read at 280, 320 and 360 nm. The analyses were performed at 24 °C and 0.7 mL/min flow rate. Samples were automatically injected at volume 10 μL. Two eluents were used in gradient mode. Eluent A was 0.1% phosphoric acid in deionized water (dH_2_O) and eluent B was acetonitrile. The steps of the gradient are described on Table 8.

The dried grape extracts from skins and seeds were dissolved in 70% acetonitrile in dH_2_O at final concentration 10 mg/mL. The skin extracts were loaded for 1 min in an ultrasonic bath for completely dissolution. After that, the samples were filtered through a 0.8 μm filter and then by 0.45 μm filter units. The content in the obtained peaks in chromatograms were proved by standard solutions: gallic acid, (+)-catechin, (−)-epicatechin, (−)-epigallocatechin gallate, procyanidin B1, procyanidin B2, procyanidin B3, procyanidin C1, oenin, quercetin 3-glucoside, resveratrol, quercetin and rutin. Neither grape seed extract nor grape skin extract have a characteristic rutin peak.

### 3.6. Data Processing of HPLC Method

All of the experiments were performed five times and average values were calculated. The mathematical operations were performed by Microsoft Excel. The standard solutions were used for the derivation of the linear equations and correlation coefficients of determination (R^2^). The limit of detection (LOD) was the lowest concentration of the compound in the sample that could be consistently detected with a stated probability. The limit of quantification (LOQ) is the lowest concentration that can be quantified accurately and precisely [48]. The used equations are:$$\mathrm{LOD}=\frac{3.3\times \mathrm{SD}}{b}$$
$$\mathrm{LOQ}=\frac{10\times \mathrm{SD}}{b}$$

“SD” is the standard deviation and “b” is the slope of the obtained curve. Additionally, the HPLC characteristic (retention time, RT) of the peaks was analyzed. The relative standard deviations (RSD) of those parameters were calculated: RSD% = (SD × 100)/$\bar{x}$.

### 3.7. Statistical Analysis

Experimental results were means ± SD of three parallel measurements. Analysis of variance was performed by ANOVA procedures (DPS 7.55 for Windows).

## 4. Conclusions

In summary, skin extracts of the studied four grape varieties are a promising medical and food source which deserve more intensive research. The values of the total phenolic contents, total flavonoids, total anthocyanins, procyanidins and ascorbic acid of the skin extracts were determined and compared with the analogical parameters of the seed extracts of the same cultivars. It was found that the investigated parameter values of the skin extracts were smaller than those of seed extracts, determined previously. According to the phenolic content and antioxidant capacity of the skin extracts, the different grape varieties were arranged by the following order: Marselan ≥ Pinot Noir > Cabernet Sauvignon > Tamyanka. The individual compounds in skin extracts were identified by HPLC. The amount of determined catechin, epicatechin and procyanidin C1 of the skin extracts were smaller than those of seed extracts. Only the concentrations of gallic acid and procyanidin B3 in skin and seed extracts were similar. The skin extracts contained additional valuable bioactive compounds like oenin, quercetin and quercetin-glucoside. It was proved that skin extracts contain the strong antioxidants procyanidin B3 and procyanidin C1. In the future, it is necessary to investigate the role and acts of the mechanism of these natural individual or combined bioactive compounds in different diseases, as well as to study the application of the obtained extracts as food additive and dietary nutrients.

## Figures and Tables

**Figure 1 molecules-28-04061-f001:**
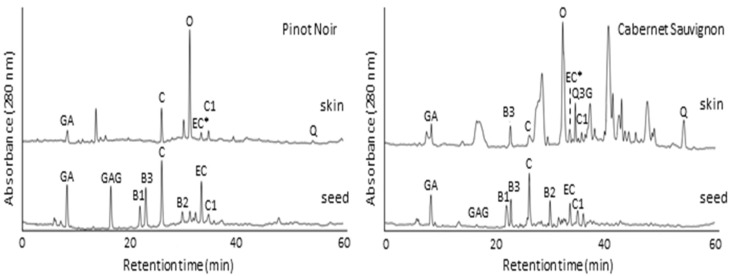
Chromatograms of skin and seed extracts from Pinot Noir and Cabernet Sauvignon red grape varieties. GA—gallic acid, C—(+)-catechin, EC*—(−)-epicatechin + (−)-epigallocatechin gallate, B1—procyanidin B1, B2—procyanidin B2, B3—procyanidin B3, C1—procyanidin C1, O—oenin, Q3G—quercetin 3-glucoside, R—resveratrol, Q—quercetin.

**Figure 2 molecules-28-04061-f002:**
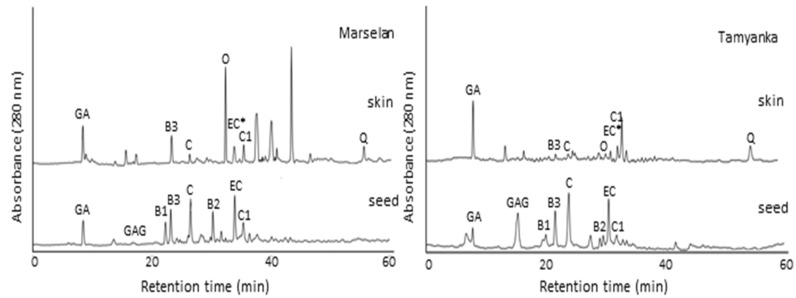
Chromatograms of skin and seed extracts from Marselan red grape and Tamyanka white grape varieties. GA—gallic acid, C—(+)-catechin, EC*—(−)-epicatechin + (−)-epigallocatechin gallate, B1—procyanidin B1, B2—procyanidin B2, B3—procyanidin B3, C1—procyanidin C1, O—oenin, Q3G—quercetin 3-glucoside, R—resveratrol, Q—quercetin.

**Table 1 molecules-28-04061-t001:** Comparison of polyphenolic content of grape skin and seed extracts.

Grape Extract	Parameters	Pinot Noir	Cabernet Sauvignon	Marselan	Tamyanka
Skin	Yeld, %	5.52 ± 0.51	13.25 ± 0.52	15.42 ± 0.55	10.02 ±0.52
	TPC, mg GAE/g DW	45.05 ± 0.85	42.32 ± 0.32	56.17 ± 0.41	36.28 ± 0.29
	TF, mg QE/g DW	4.41 ± 0.12	6.45 ± 0.12	6.70 ± 0.16	2.64 ± 0.11
	PC, mg CE/g DW	3.31 ± 0.15	3.65 ± 0.13	4.52 ± 0.14	1.23 ± 0.10
	TA, mg CGE/g DW	1.21 ± 0.10	3.34 ± 0.12	3.94 ± 0.15	0.015 ± 0.08
	AA, mg/g DW	3.17 ± 0.16	2.72 ± 0.11	2.76 ± 0.11	3.53 ± 0.14
Seed	Yield, %	12.05 ± 0.77	16.02 ± 0.65	18.34 ± 0.66	15.13 ± 0.63
	TPC, mg GAE/g DW	111.22 ± 1.28	88.22 ± 0.72	103.24 ± 1.11	79.06 ± 0.65
	TF, mg QE/g DW	51.50 ± 0.30	45.95 ± 0.14	52.01 ± 0.34	40.05 ± 0.18
	PC, mg CE/g DW	170.45 ± 2.52	157.22 ± 2.10	152.18 ± 2.05	31.44 ± 0.23
	TA, mg CGE/g DW	0.04 ± 0.02	0.05 ± 0.02	0.062 ± 0.01	no
	AA, mg/g DW	11.07 ± 0.25	3.01 ± 0.11	2.71 ± 0.13	4.88 ± 0.13

Bar indicates Mean ± SD (*n* = 3).

**Table 2 molecules-28-04061-t002:** Calibration curves and validation data of the HPLC method used for determination of polyphenolic compounds in skin extracts.

Standard Compounds	Wave Length, nm	Conc. Range, mg/L	Equations	R^2^	LOD mg/L	LOQ mg/L	RSD %RT (*n* = 5)
GA	280	1.0–10.0	y = 19.497x − 0.5612	0.9999	0.25	0.75	0.14
(+)-C	280	10.0–50.0	y = 2.202x + 0.8166	0.9992	2.59	7.84	0.27
(−)-EC	280	10.0–100.0	y = 1.732x + 4.7597	0.9995	4.80	14.55	0.17
(−)-EGCG	280	2.0–10.0	y = 12.648x + 5.035	0.9991	0.98	2.99	0.18
B1	280	10.0–30.0	y = 2.534x − 13.617	0.9986	1.75	5.30	0.26
B2	280	10.0–20.0	y = 3.901x − 9.72	0.9966	4.32	13.10	0.23
B3	280	2.5–100.0	y = 9.254x + 1.58	0.9998	0.26	0.78	0.26
C1	280	2.0–10.0	y = 9.490x + 0.81	0.9999	0.22	0.66	0.18
O	280	1.0–5.9	y = 27.966x − 24.586	0.9991	0.31	0.94	0.46
Q3G	360	1.9–8.0	y = 11.416x − 13.243	0.9999	0.13	0.38	0.50
R	320	0.7–5.6	y = 4.928x − 2.062	0.9979	0.40	1.23	0.11
Q	360	0.7–5.6	y = 15.663x − 8.784	0.9993	0.22	0.65	0.79

GA—gallic acid, (+)-C—(+)-catechin, (−)-EC—(−)-epicatechin, (−)-EGCG—(−)-epigallocatechin gallate, B1—procyanidin B1, B2—procyanidin B2, B3—procyanidin B3, C1—procyanidin C1, O—oenin, Q3G—quercetin 3-glucoside, R—resveratrol, Q—quercetin.

**Table 3 molecules-28-04061-t003:** Polyphenols in grape skin and seed extracts in mg/g, expressed by dry weight (*n* = 5).

Grape Extract	Polyphenols	Pinot Noir	Cabernet Sauvignon	Marselan	Tamyanka
Skin	Gallic acid	0.30 ± 0.12	0.37 ± 0.13	0.65 ± 0.18	1.11 ± 0.54
	Procyanidin B3	-	0.65 ± 0.19	0.97 ± 0.23	0.39 ± 0.11
	(+)-Catechin	5.86 ± 0.97	1.51 ± 0.69	2.23 ± 0.61	0.44 ± 0.14
	Oenin	2.21 ± 0.63	2.66 ± 0.85	2.54 ± 0.71	0.99 ± 0.31
	(−)-EC*(Epicatechin + Epigallocatechin gallate)	0.45 ± 0.15	0.54 ± 0.13	0.47 ± 0.15	0.21 ± 0.09
	Quercetin 3-gallate	-	1.84 ± 0.78	-	-
	Procyanidin C1	0.32 ± 0.12	0.39 ± 0.14	0.42 ± 0.14	0.12 ± 0.08
	Quercetin	0.67 ± 0.34	1.25 ± 0.37	1.15 ± 0.55	0.45 ± 0.15
Seed	Gallic acid	0.61 ± 0.23	0.44 ± 0.30	0.42 ± 0.31	0.35 ± 0.15
	Gallic acid glucoside	0.88 ± 0.32	0.13 ± 0.07	0.05 ± 0.01	0.64 ± 0.02
	Proanthocyanidin B1	8.81 ± 1.09	8.82 ± 1.17	7.57 ± 0.65	7.05 ± 0.73
	Proanthocyanidin B3	2.90 ± 0.39	0.95 ± 0.37	1.31 ± 0.93	1.69 ± 0.73
	(+)-Catechin	12.16 ± 0.98	9.17 ± 0.73	8.06 ± 0.51	7.35 ± 0.50
	Proanthocyanidin B2	4.06 ± 0.41	5.17 ± 0.21	5.17 ± 0.99	3.46 ± 0.62
	(−)-Epicatechin	10.16 ± 1.09	5.94 ± 0.45	14.27 ± 0.64	4.89 ± 0.31
	Proanthocyanidin C1	0.34 ± 0.11	0.55 ± 0.21	0.68 ± 0.32	0.14 ± 0.07

Bar indicates Mean ± SD (*n* = 5).

**Table 4 molecules-28-04061-t004:** Retention time (RT) and relative standard deviation (%RSD) of analyzed compounds in the tested grape skin and seed extracts by HPLC.

Compounds in Exstracts	Retention Time of Four Grape Extracts, min	%RSD, Pinot Noir Extract	%RSD, Cabernet Sauvignon Extract	%RSD, Marselan Extract	%RSD, Tamyanka Extract	%RSD of Four Grape Extracts
GA	8.25	0.18	0.54	0.48	0.06	1.77
GAG	16.58	0.16	0.25	0.13	0.14	2.71
(+)-C	25.67	0.35	0.09	0.04	0.04	3.81
(−)-EC	33.16	0.09	0.04	0.13	1.64	3.29
B1	22.13	0.10	0.25	0.23	0.41	1.13
B2	29.96	0.08	0.40	0.11	0.32	0.79
B3	23.04	0.05	0.17	0.19	0.31	0.68
C1	34.60	0.0	1.42	0.20	1.70	3.64
O	31.68	0.08	0.11	0.33	0.08	3.14
Q3G	34.88	-	0.22	-	-	0.22
Q	54.86	0.02	0.09	0.49	0.36	1.53

**Table 5 molecules-28-04061-t005:** Identification of compounds in the tested Marselan grape skin and seed extracts by HPLC, using standard addition.

Compounds in Extracts	Standard Addition, μg/mL	Absorbance Ratio, (Ax + Ast)/Ast	Standard Addition, μg/mL	Absorbance Ratio, (Ax + Ast)/Ast
GA	5	1.70	10	1.58
(+)-C	20	1.23	40	1.13
(−)-EC	45	1.37	90	1.23
B1	12	1.09	24	1.05
B2	10	1.69	15	1.59
B3	17	1.70	35	1.52
C1	5	1.44	10	1.28
O	2	1.95	4	1.90
Q3G	3	1.99	6	1.98
Q	2	1.93	4	1.88

Ax—absorbance of compound in samples, Ast—absorbance of standard addition.

**Table 6 molecules-28-04061-t006:** Comparison of flavonols content and quercetin of tested skin extracts with results from other studies.

Grape Skins	PC B1, mg/g	PC B2, mg/g	PC B3, mg/g	PC C1, mg/g	C, mg/g	EC, mg/g	Q, mg/g	Reference
Cabernet Sauvignon	0.012 **	0.001 **	0.027 **	-	0.017 **	0.006 **	0.048 **	[27]
Merlot	0.002 **	0.021 **	0.035 **	-	0.025 **	0.013 **	0.031 **	[27]
Cabernet Sauvignon	-	-	-	-	1.41	1.27	0.006	[38]
Cabernet Sauvignon	-	-	-	-	-	-	0.23	[32]
Grenache	0.62	0.41	0.27	0.46	0.76	0.28	-	[25]
Carignan	0.74	0.43	0.28	0.53	1.01	0.35	-	[25]
Pinot Noir	-	-	-	-	0.13	-	0.15	[32]
Pinot Noir	0.30	0.28	-	-	0.61	0.34	1.54	[28]
Pinot Noir	-	-	-	0.32	5.86	0.45 *	0.67	Current study
Cabernet Sauvignon	-	-	0.65	0.39	1.51	0.54 *	1.25	Current study
Marselan	-	-	0.97	0.42	2.23	0.47 *	1.15	Current study
Tamyanka	-	-	0.39	0.12	0.44	0.21 *	0.45	Current study

All results were presented as mg/g dry weight extract, only data marked with **—mg/g fresh grape. EC—(−)-epicatechin, the data marked with * are combined results (−)-epicatechin (EC) + (−)-epigallocatechin gallate (EGCG). C—(+)-catechin, B1—procyanidin B1, B2—procyanidin B2, B3—procyanidin B3, C1—procyanidin C1, Q—quercetin.

**Table 7 molecules-28-04061-t007:** Spectrophotometric determination of antioxidant capacity of grape skin and seed extracts.

Grape Extract	Parameters	Pinot Noir	Cabernet Sauvignon	Marselan	Tamyanka
skin	DPPH, µM TE/g DW	75.77 ± 1.12	81.23 ± 0.73	89.74 ± 0.78	14.22 ± 0.18
	ABTS, µM TE/g DW	87.61 ± 1.25	99.05 ± 0.88	109.31 ±1.01	58.23 ± 0.41
	FRAP, µM TE/g DW	11.28 ± 0.29	35.68 ± 0.45	29.05 ± 0.28	10.03 ± 0.34
	CUPRAC, µM TE/g	12.53 ± 0.32	14.41 ± 0.17	18.93 ± 0.22	11.64 ± 0.18
seed	DPPH, µM TE/g DW	579.33 ± 4.15	435.25 ± 3.3	597.23 ± 4.12	245.6 ± 3.23
	ABTS, µM TE/g DW	2203.51± 10.25	2246.23 ± 11.33	2273.92 ± 12.32	1907.24 ± 9.56
	FRAP, µM TE/g DW	142.76 ± 2.35	97.15 ± 0.82	115.22 ± 1.10	67.45 ± 0.75
	CUPRAC, µM TE/g	68.18 ± 0.98	64.38 ± 0.52	72.34 ± 0.81	32.23 ± 0.29

Bar indicates Mean ± SD (*n* = 3).

**Table 8 molecules-28-04061-t008:** Gradient mode of RP-HPLC for grape extracts analyses.

Time, min	Eluent A, %	Eluent B, %
0.0–5.0	90	10
5.1–20.0	80	20
20.1–25.0	75	25
25.1–50.0	70	30
50.1–80.0	55	45

## Data Availability

Not applicable.

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
