# Peer review of "Comparison of Identification and Quantification of Polyphenolic Compounds in Skins and Seeds of Four Grape Varieties"

_molecules, 2023, doi:10.3390/molecules28104061_

Round 1

Reviewer 1 Report

Comparison of identification and quantification of polyphe- nolic compounds in skins and seeds of four grape varieties

General Comments:

1. Once Molecules has a broad scope, I feel there are too many abbreviations/acronyms in the Abstract. Not all readers are acquainted with them, and the Abstract should provide a complete view of the research without the need to consult all details.

I suggest the authors rephrase the Abstract to reduce/eliminate the abbreviations.

2. Lines 98-99

...” nol (70%, v/v) was chosen to carry out the extraction, because ethanol is harmless to health...”

I suggest the authors rephrase this statement once ethanol is not harmless to health if one drinks it frequently, at high concentrations. It is a substantial international health problem due to increased alcoholism worldwide in the last decade. So, the generalization that … "ethanol is harmless to health...” should be carefully revised.

3. Although I consider the subject relevant and that the authors developed a good work, there is a point that hinders me to recommend the work in its present form for publication in Molecules.

In line 79, the authors state:

…”. The present work addresses the flavanol composition of grape skins, comparing the results with those found from the seeds of four grape varieties; the latter information has been published in an earlier work [33]…”

33. Krasteva, D.; Ivanov, Y.; Chengolova, Z.; Godjevargova, T. Antimicrobial Potential, Antioxidant Activity, and Phenolic Content of Grape Seed Extracts from Four Grape Varieties. Microorganisms 2023, 11, 395. 543 https://doi.org/10.3390/ microorganisms11020395,

So, the chromatographic aspects involving the “identification" and quantifications of the grape seed extract that are compared in the present work were done in the previous report (ref 33)

[Figure S1. RP-HPLC analyses of grape seed extract with absorbance detection at 280 nm: GA – gallic acid, GAG – gallic acid glucoside, B1 – Procyanidin B1, B3 – Procyanidin B3, (+)-C – catechin, B2 – Procyanidin B2, (-)-EC – epicatechin, C1 – Procyanidin C1.]

Unfortunately, although the work is already published, I see a series of problems involving the chromatographic analysis in both works (previous one, ref 33 and this one). The main issues are:

a. As mentioned by the authors, grape extracts (seeds, skin, …) are complex extracts containing many compounds. This is clearly exemplified by the authors in Fig. 1 of the present work, which display the peaks corresponding to compounds that absorb in 280 nm. Therefore, the entire chromatogram using either a Photodiode array UV-VIS detector or a mass spectrometer would show the complexity of this kind of sample. So, the "identification" of the selected compounds using the absorbance at 280 nm is not acceptable for analyzing such complex samples nowadays.

When mass spectrometry is not available, at least a systematic study using the Absorbance ratio at two different wavelengths (for instance, 280/220 nm) followed by a retention index system based preferentially on two columns of different characteristics would give some more robust information about the "identity" of the peak under investigation.

b. Mass spectrometry (if possible HRMS or at least tandem MS/MS) would give more trustable information on the "peaks identity".

c. It is easy to see that the retention time of some compounds does not repeat from sample to sample, although the RSD within the sample could be acceptable. For instance, the present work's peak C1 in Figures 1 and 2 does not maintain its retention time from sample to sample. Although the retention time axis does not exhibit the marks for each number corresponding to each 10 minutes interval, it is easy to observe the absence of repeatability. So, although the scale supplied is not precise to determine the precise retention time of each compound, it is possible to expand the chromatogram and draw reasonably precise (for the purpose) axis information. Take just peak C1 as an example:

Figure 1: in the chromatogram corresponding to the Pinot Noir seed, the retention time of C1 is close to 35 min. However, in the same figure, the retention time of C1 in Cabernet Sauvignon seed is much higher than that. 

Figure 2: the retention time of the same C1 in the Tamyanka seed is much closer than 30 minutes.

So, comparing the retention time of Cabernet Sauvignon seed (Fig. 1) with Tamyanka seed (Fig. 2), a large discrepancy in retention time is observed. Considering the high quality of the current pumps, columns, and software, this large discrepancy does not allow us to attribute the same identity to the two peaks. Unless the chromatographic system presents problems, and in this case, it cannot be used for "identification" purposes.

A similar problem can be seen in ref 33, which was used as a guide for the chromatographic aspects of this work.

At least a table containing the retention times of each selected compound and an RSD of several measurements would be advisable to avoid "identification" mistakes.

-

Reviewer 2 Report

In the paper titled “Comparison of identification and quantification of polyphenolic compounds in skins and seeds of four grape varieties”the authors presented well done work regarding the polyphenolic profile of seed and skin extracts.

The analytical methods were well described and appropriate. The results and discussion are clearly presented. I recommend accepting the manuscript after minor revisions regarding the text editing- the authors should revise the text for mispellings.

The paper is well written, but needs some spelling check.

Reviewer 3 Report

I believe that this work does have some importance, because of the bioactive properties of phenolic compounds. And, the ability to quantify these extracts will lead to identification of grape products with health benefits. My suggestion would be to highlight, just a bit more, the quality of the extracted phenolic compounds. I think this would highlight the importance of this manuscript. 

I believe that the HPLC work was done well. I’m adding “Accept in present form”. However, one last look for grammar wouldn’t be an issue.

Round 2

Reviewer 1 Report

1. The authors have improved the manuscript to a condition that it is now "publishable".

2. I do not agree with the authors that just "spiking" (adding a solution containing an analytical standard of the compounds under study) and having a peak increment means that the injected compound is the same as the one investigated (retention time is characteristic but not unique).

The English Language does not hinders the readers to understand the content of the manuscript. Small gramar mistakes are still present in the MS; if accepted I suggest a language review before publication.